# Management of Brain Metastases from Human Epidermal Growth Factor Receptor 2 Positive (HER2+) Breast Cancer

**DOI:** 10.3390/cancers14205136

**Published:** 2022-10-20

**Authors:** Tresa M. McGranahan, Alipi V. Bonm, Jennifer M. Specht, Vyshak Venur, Simon S. Lo

**Affiliations:** 1Department of Neurology, Alvord Brain Tumor Center, University of Washington School of Medicine, Seattle, WA 98195, USA; 2Virginia Mason Franciscan Health, Seattle, WA 98101, USA; 3Division of Medical Oncology, Fred Hutchinson Cancer Center/University of Washington, Seattle, WA 98109, USA; 4Department of Radiation Oncology, University of Washington School of Medicine, Seattle, WA 98195, USA

**Keywords:** breast cancer, brain metastasis, Human Epidermal growth factor Receptor 2

## Abstract

**Simple Summary:**

Treatment options for patients with Human Epidermal growth factor Receptor 2 positive (HER2+) metastatic breast cancer are rapidly changing, especially for patients with brain metastasis. Historically, treatment options for brain metastasis were focused on local therapies, radiation and surgery. There are now multiple targeted therapies that are able to treat brain metastasis and prolong the lives of patients with HER2+ breast cancer. With the growing number of treatment options, making medical decisions for patients and clinicians is more complicated. This paper reviews the treatment options for patients with HER2+ breast cancer brain metastasis and provides a simplified algorithm for when to consider delaying local treatments.

**Abstract:**

In the past 5 years, the treatment options available to patients with HER2+ breast cancer brain metastasis (BCBM) have expanded. The longer survival of patients with HER2+ BCBM renders understanding the toxicities of local therapies even more important to consider. After reviewing the available literature for HER2 targeted systemic therapies as well as local therapies, we present a simplified algorithm for when to prioritize systemic therapies over local therapies in patients with HER2+ BCBM.

## 1. Introduction

Brain metastases (BMs) from solid tumors have classically been approached in a tissue agnostic manner with a focus on local therapy options. The presence of a BM has historically been considered an exclusion criterion for most systemic clinical trials due to the perception that BMs confer poor prognosis. Data from more modern cohorts challenge this notion, driven both by the availability of targeted and brain-penetrant therapies, and also by the improvement in magnetic resonance imaging (MRI) technology, which has increased early detection of small and clinically asymptomatic brain metastases. This led to the inclusion of patients with untreated and asymptomatic brain metastases in registration trials for novel therapies including tucatinib and trastuzumab deruxtecan (T-DXd). As a result, over the past three years, there is growing evidence of the radiographic and symptomatic response of breast cancer brain metastasis (BCBM) to systemic agents. The options of systemic therapy for treatment of BCBM have raised the question of when to use local therapy in HER2+ BCBM. This paper reviews the available literature for systemic agents for HER2+ BCBM, evidence and toxicities of local therapy and discussion of clinical situations where delaying local therapy may be favored. 

## 2. Incidence of Breast Cancer Brain Metastasis

Determining the true incidence of BCBM is associated with our ability to detect BCBM. Early autopsy studies predating the clinical use of the MRI found the incidence of BCBM to be 30%. However, only 31% of pathologically confirmed BCBM cases had known clinical involvement during life and only 14% were felt to be the immediate cause of death [1]. Since that time, MRI technology has advanced, so that small asymptomatic BCBM are often identified incidentally during imaging for other reasons.

The incidence of BCBM varies by subtype as reported in a recent meta-analysis of articles published between 2000–2020, with the pooled cumulative incidence of BM being 31% for HER2+ and 32% for triple negative subgroup, while the hormone receptor (HR) was notably lower at 15% [2]. For HER2+ metastatic breast cancer (mBC), this is a 13% incidence of BM per patient year. For patients with mBC and germline BRCA1/BRCA2 mutations, the incidence of BM has been reported to be as high as 67% [3,4].

Risk factors for BCBM established by The International Breast Cancer Study Group (IBCSG) include the presence of lung metastasis, HER2+ disease, hormone receptor (HR) negative disease and age < 35 [5]. While multiple studies have suggested the need for central nervous system (CNS) surveillance imaging, at this time, the IBCSG and the American Society of Clinical Oncology (ASCO) guidelines state that there is not sufficient evidence to support routine brain screening for asymptomatic BMs in any subgroup [6]. While screening for asymptomatic BCBM is not standard, once BMs are identified, the presence of even asymptomatic BCBM often guides the selection of therapeutic strategies.

In parallel with the increasing incidence of BCBM, the overall prognosis is improving in patients with HER2+ BCBM. Improved prognosis is likely driven by the targeted systemic agents as highlighted by a retrospective series of 123 patients with HER2+ BCBM divided patients into three cohorts by treatment years corresponding to the development of trastuzumab (1998–2007), lapatinib (2008–2012), and pertuzumab (2013–2015). Median survival improved from 3.6 to 6.6 to 7.6 years in successive cohorts [7]. In this same study, patients with BCBM who received HER2 directed therapy had a median overall survival (mOS) of 2.1 years, compared with 0.65 years in those who did not. Concordantly, recent SEER data suggest that patients with HER2+ BCBM and selected clinical features have 2-year overall survival rates exceeding 60% [8]. It is notable that these available retrospective data precede the clinical availability of data for trastuzumab deruxtecan (T-DXd) and tucatinib. Given this increasing survival, consideration of long-term toxicity with local therapy for BCBM becomes paramount.

## 3. HER2-Targeted Systemic Agents with Data to Support CNS Activity

Currently, RANO-BM is the preferred criteria for evaluating the response of a BM and defines a measurable disease as greater than or equal to 10 mm [9]. Using the sum of the longest diameter of up to 5 target BM, RANO-BM defines a partial response (PR) as a decrease in this total diameter of 30% or greater. Protocols written prior to the development of RANO-BM defined the response with the variable decrease in diameter from 20% to 50%. As a result, the objective responses rate (ORR) is not standardized across older BCBM studies. Regardless of the percentage of the response, across multiple studies of patients with BCBM, the ORR does correlate with the median progression free survival (mPFS) and the mOS compared to non-responders in Table 1. In symptomatic patients, response may be associated with improved neurologic symptoms. While the ORR is an important objective, clinical benefit is seen with treatments that slow further progression of CNS disease, so evaluating the disease control rate (DCR) inclusive of complete response (CR), PR and stable disease (SD) are also reported for the clinical trials reviewed in Table 1.

To date, there is wide variation in prospective studies of BCBM. Historically, only patients with BCBM treated with definitive radiation therapy (XRT) that were asymptomatic and stable on follow-up imaging were allowed in clinical trials. More recently, patients with asymptomatic, untreated, small BCBM or BCBM that have asymptomatic progression despite XRT have been included. Given the narrow scope of patients enrolled in clinical trials, the recently updated ASCO guidelines for management of HER2+ BCBM do not recommend trials of systemic therapy for newly diagnosed symptomatic BCBM [6]. In clinical trials that have included symptomatic patients, as well as in clinical experience, patients can have dramatic improvement in neurologic symptoms with HER2+ targeted systemic agents that have CNS activity (Figure 1) [13,15,24,25]. There are no head-to-head comparisons of local therapy compared to systemic therapy making decisions in this rapidly evolving space complicated for patients and providers.

### 3.1. Trastuzumab

Trastuzumab is a monoclonal antibody against the extracellular domain of HER2, which received initial approval in 1998 as the first HER2-targeted therapy. Unfortunately, standard dosing of trastuzumab has limited ability to cross the blood–brain barrier (BBB) as demonstrated by a study of patients with paired serum and cerebrospinal fluid (CSF) samples. The study collected CSF and serum samples before or after radiation and found that the baseline CSF:serum ratio of trastuzumab is only 1:420, which only improved to 1:49 after radiation [26]. Despite limited CNS concentrations, inclusion of trastuzumab is associated with a longer time to CNS metastasis [27].

Recently, the phase 2 PATRICIA study tested high-dose trastuzumab (6 mg/kg intravenous weekly) with pertuzumab, a monoclonal antibody that blocks HER2 dimerization (also administered intravenously), in patients with progressive BCBM. In 39 patients, there were 4 PR and CNS ORR was 11% [17]. Notably, the 6-month DCR was 51%, and two patients had stable disease for over 2 years.

### 3.2. Lapatinib

Lapatinib is a first-generation tyrosine kinase inhibitor (TKI) that acts as a dual-inhibitor of HER1/HER2 with the ability to cross the BBB. In patients with progressive BCBM following prior radiation, lapatinib monotherapy resulted in a 20% or greater decrease in diameter of BCBM in 21% of patients (only 6% when the response was defined as a 50% or greater reduction in longest diameter) [18]. When the protocol was amended to allow 50 patients to be treated with the combination of lapatinib with capecitabine, 20% of patients had a reduction in the size of the BCBM of 50% or greater and 40% had a response of 20% or greater [18]. The activity of capecitabine–lapatinib combination for BCBM was further supported by the phase 2 LANDSCAPE trial of 45 patients where the intracranial ORR was 65.9% (defined as a 30% reduction) and the CNS mPFS was 5.5 months [19]. Notably, of the 24 patients with neurologic symptoms at baseline, 58% had improvement in neurologic symptoms with treatment and 67% had an objective response.

### 3.3. Neratinib

Neratinib is a second-generation TKI which acts as an irreversible pan-HER TKI with activity in BCBM, even in patient who were previously treated with lapatinib. The TBCRC022 trial evaluated both patients previously treated with lapatinib and patients who were lapatinib naïve with BCBM that were progressive despite radiation. In 37 lapatinib-naïve patients with HER2+ BCBM, neratinib-capecitabine resulted in an ORR of 49% and a mPFS of 5.5 months, signifying intracranial activity [20]. It is notable that while the lapatinib treated cohort closed for slow accrual, there was an ORR of 33% in patients previously treated with lapatinib. Based on these results, neratinib received an orphan drug designation for HER2+ BCBM in 2019.

Subsequently, the phase 3 NALA study compared neratinib-capecitabine to lapatinib-capecitabine in metastatic HER2+ breast cancer in patients who had progressed on two or more HER2+ directed therapies [24]. The study included patients with BCBM, but baseline scans were not mandated, limiting the ability to interpret the results. The authors reported that 22.8% of patients treated with neratinib-capecitabine required intervention for CNS metastases, compared with 29.2% for lapatinib-capecitabine [24]. Neratinib also binds to the binding pocket of p-glycoprotein ABCB1 and reduces drug efflux; thus, the combination with chemotherapies may be favorable [28]. In a randomized trial of 479 patients with recurrent or mBC that included 18 patients with stable, treated BCBM, a key finding was that treatment with neratinib and paclitaxel was associated with a lower risk (RR = 0.48) of CNS recurrence and a delayed time to BM compared to trastuzumab and paclitaxel [29].

### 3.4. Tucatinib

Tucatinib, a third-generation oral TKI that is selective for the kinase domain of HER2, was FDA approved in 2020. The registration phase 3 HER2CLIMB trial studied the addition of tucatinib or placebo to trastuzumab–capecitabine in patients with metastatic HER2+ breast cancer and had a prespecified endpoint of activity in HER2+ BCBM [15]. The study not only allowed patients with treated BCBM, but also patients with untreated BCBM unless they were in need of immediate local intervention. A total of 291 patients with BCBM were included in the study, and in these patients 1 year CNS mPFS was 35% with tucatinib and 0% in the placebo group. Estimated 1-year survival was 71.7% with tucatinib and 41.1% with placebo, trastuzumab–capecitabine. In the 75 patients with measurable intracranial disease, the CNS ORR was 47% with the addition of tucatinib and 20% with trastuzumab–capecitabine alone. Additionally, in a subset of patients who enrolled with untreated BCBM and deferred radiation, the CNS mPFS was 8.1 months with the addition of tucatinib compared to 3.1 months with placebo, trastuzumab–capecitabine. A subsequent patient-reported outcomes analysis from HER2CLIMB reported that deterioration of quality of life was substantially reduced in patients treated with tucatinib [25].

### 3.5. Trastuzumab Emtansine

Trastuzumab emtansine (T-DM1) was the first antibody-drug conjugate (ADC) approved in second line for treatment of metastatic HER2+ breast cancer after progression on trastuzumab and pertuzumab, based on the EMILIA trial [30]. This trial excluded patients with untreated or progressive BCBM. Subsequently, subgroup analysis of 398 patients with BCBM enrolled in the phase IIIb open label single arm study has reported an intracranial ORR of 21.4% and DCR of 42.9%, confirming the CNS activity of T-DM1 [16]. Given the evidence of the CNS efficacy of tucatinib in the HER2CLIMB trial, the ongoing HER2CLIMB-02 trial includes patients with untreated and progressive BMs and will compare T-DM1 with and without tucatinib in the metastatic setting (NCT03975647).

### 3.6. Trastuzumab Deruxtecan

Trastuzumab deruxtecan (T-DXd) is the second ADC to be approved for metastatic HER2+ breast cancer but differs from T-DM1 due to a cleavable topoisomerase I inhibitor intended to act on both HER2+ and nearby cells. T-DXd was first approved in 2019 as a third-line therapy for HER2+ mBC based on results from the phase 2 DESTINY-breast01 trial [31]. However, the Destiny-Breast03 trial moved T-DXd to the preferred 2nd line therapy after the study of 524 patients with mBC were randomize to T-DM1 or T-DXd. This study reported improvement in the ORR (79.7% in patients treated with T-DXd compared to 34.2% for T-DM1) and in 12-month PFS (75.8% for patients receiving T-DXd compared to 34.1% with T-DM1) [32]. This study allowed 82 patients with clinically stable, previously treated BCBM and found a mPFS of 15 months with T-DXd compared to 3 months with T-DM1 and stable BCBM in 23.8% of patients treated with T-DXd compared to 19.8% treated with T-DM1[13]. In no other study has the mPFS exceeded a year for patients with HER2+ BCBM (Table 1).

The future use of T-DXd is expanding with the results of the DESTINY-Breast04 trial, which evaluated survival of patients with mBC with low levels of HER2 (IHC1+ or IHC2+ with negative FISH) treated with T-DXd compared to treating physician’s choice chemotherapy (TPC). In patients with HR+ HER2 low mBC, the OS was 23.9 months for T-DXd vs. 17.5 months for TPC, and in the overall cohort it was 23.4 months vs. 16.8 months. Similarly, the mPFS was improved with T-DXd at 10.1 vs. 5.4 months in HR+ patients, and similarly improved in the overall cohort at 9.9 vs. 5.1 months [33]. Of the 557 patients in the study, a total of 32 had stable, treated brain metastases, and the outcomes for this cohort are eagerly awaited.

Two trials have now specifically studied the utility of T-DXd in BCBM. The ongoing five-cohort DEBBRAH trial recently reported interim results for HER2+ BCBM, and the intracranial ORR for asymptomatic, untreated BCBMs was 50%, whereas for progressing BCBMs it was 44% [13]. Perhaps equally important to the response rate, patient’s quality of life (QOL) was maintained at 6 months. Further data from this study regarding the response in HER2+ compared to HER2- low expressing and in patients with leptomeningeal disease have yet to be reported.

Similarly, the prospective single arm phase 2 TUXEDO-1 trial evaluated T-DXd activity in HER2+ patients with active brain metastases. The first 15 patients have been reported with an intracranial response rate of 73.3% and a mPFS of 14 months [12].

There are two ongoing studies evaluating T-DXd in HER2+ stable or progressing BCBMs that will further contribute to understanding of the role in management of BCBM. The DESTINY-Breast07 is a phase 1b/2 trial evaluating T-DXd in combinations with durvalumab, pertuzumab, paclitaxel, or tucatinib for patients with stable, treated brain metastasis. For patients with active brain metastasis, there are two arms evaluating T-DXd in combination with tucatinib as well as monotherapy. Additionally, the phase 3b/4 DESTINY-Breast12 is evaluating survival as well as QOL and cognitive function of T-DXd in patients with or without BCBM.

## 4. Local Treatment

### 4.1. Surgery

The role of surgery in the treatment of BMs is limited to patients with (1) a single large BM, (2) a need for symptom control and/or (3) a need for diagnosis. The data guiding the role in surgery in BMs come from studies of mixed populations of BMs as there have been no prospective studies examining the role of surgery in BCBMs.

In 1990, Patchell published the fundamental study supporting survival benefits to surgical resection in patients with a single large BM [34]. This study randomized 48 patients with a single BM to WBRT either with or without surgical resection. Patients who underwent surgical resection survived 40 weeks compared to just 15 weeks in the WBRT-only patients. Not only did surgery prolong survival, but patients in the surgical arm also maintained functional status 30 weeks longer than patients in the WBRT arm and had a lower rate of local recurrence. The benefits to surgery for prolonging both survival and functional status in patients with a single BM was also found in a second prospective randomized study that also found the benefit was greatest in patients with stable extracranial disease where the median OS with surgery was 12 months compared to 7 months without surgery [35].

It is notable that both of these prospective studies were before the use of SRS, modern MRI and chemotherapies with activity in BMs. There is, however, a multicenter retrospective study of 223 patients with a larger (defined as greater than or equal to 2 cm in diameter) BM treated with either single fraction stereotactic radiosurgery (SRS) alone or with surgery and SRS [36]. This study included 22% of patients with BCBM and found that despite the surgical group having larger tumor volumes, the patients treated with both surgery and SRS had a longer survival of 15.2 months compared to 10 months for patients treated with SRS alone. Similar to the WBRT data, the 1-year rate of local recurrence was 20.5% in the surgery + SRS arm and 36.7% in the SRS alone. Based on retrospective data, prognostic factors for surgically resected single-brain metastasis include age less than 65, non-small cell lung cancer histology, use of radiosurgery and control of extracranial disease [37]. Collectively, these data support an important role for surgery prolonging survival in patients with single large brain metastasis where a gross total resection is possible.

While there is only level 1 evidence for surgical resection in single large BM, there are multiple surgical case series that report for patients where all BMs are surgically removed; survival is comparable to patients who underwent surgical resection for a solitary brain metastasis [38,39,40]. Survival in these patients is also improved compared to patients who had surgical resection with residual disease (5.8–6 months) or a removed BM (10.6–14 months) [39,40].

Clinically, surgical removal of a BM is often considered when a patient’s functional status may be improved with surgical intervention. Surgery may provide symptomatic benefit for patients with a large (>2.5 cm diameter) symptomatic lesion in a surgically amenable location. An example of a surgically amenable location could be an intraventricular metastasis that causes obstructive hydrocephalus, and resection of the mass would allow for treatment of intracranial pressure. Another example may be a large frontal lobe metastasis where resection of the mass could improve hemiplegia by relieving the mass effect. In these clinical scenarios, surgery may reduce the length of time that patients require steroid therapy and may improve a patient’s functional status so that the patient may be eligible for systemic therapies post-operatively. While it is unlikely that a prospective study would be able to capture these complex clinical scenarios, a retrospective study of 750 patients, of which 15% had BCBM, did find that functional status was significantly improved by surgical resection with an increase in median KPS from 80 preoperatively to 90 post-resection [41].

Additionally, surgery may play an important diagnostic role in patients with BCBM. Some patients present with life threatening mass effect from BMs without a known primary cancer diagnosis. In this setting, surgical removal of the symptomatic BM provides diagnosis as well as symptomatic benefit even in the setting of multiple BMs. Additionally, in the setting of progressive symptomatic or radiographic changes following radiation, surgery can provide relief of symptoms as well as clarify if the changes represent recurrent disease or radiation necrosis.

However, the above benefits to surgical resection of BCBM must be balanced with the risks of surgery. Venous thromboembolic events (VTE) are the most reported complication following surgery for a BM [42]. While less common, complications such as wound healing and infection can lead to delays in radiation therapy or systemic therapy and should be considered when evaluating the role of surgery in treatment of BCBM. Especially for surgery in eloquent brain areas, the need for prolonged rehabilitation following surgery should be balanced with overall life expectancy in patients with BCBM.

In addition to postoperative complications, surgery for BCBM has also been associated with an increased risk of leptomeningeal metastasis (LM) in multiple retrospective studies [43,44]. The rates of LM are higher with surgery in the posterior fossa as well as with piecemeal resection technique. En-bloc surgical resection of BCBM should always be preferred given that there are both lower rates of local recurrence and LMD with en-bloc resection [36,45]. Collectively, when possible, we favor the use of surgery for patients where en-bloc GTR of a symptomatic BCBM is likely to improve neurologic function faster than treatment with either radiation or systemic therapy.

### 4.2. Stereotactic Radiosurgery (SRS) and Stereotactic Radiotherapy (SRT)

SRS has become the cornerstone of local therapy for BCBM, particularly given that BCBMs are sensitive to radiation. While guidelines support the use of SRS alone for 1 to 4 BMs, given improved technology, SRS is often considered for patients with up to 10 small BMs. The safety of this is supported by Yamamoto et al. who enrolled 12% BCBM [46].

In multiple randomized phase III trials, WBRT in addition to SRS was associated with improved local control but not overall survival or functional independence compared to SRS alone [47,48,49,50]. However, the combined approach was associated with higher rates of cognitive deterioration, most notably immediate and delayed memory and lower QOL [48]. While cognitive testing following SRS was largely stable, QOL, specifically physical well-being, did decline in the SRS group. Li et al. reported the preliminary results of a randomized phase 3 trial comparing SRS and WBRT for patients with 4–15 metastases. SRS was associated with a reduced risk of neurocognitive deterioration compared to WBRT and there was no difference in overall survival [51]. There is an ongoing study, NCT03075072, comparing SRS for patients with 5–20 metastasis to WBRT, and it is evaluating cognitive outcomes in addition to survival.

SRS is often used in combination with the surgical removal of a BM to improve local control. While adjuvant SRS has historically been considered the standard [52,53], multi-institutional retrospective analysis has found that while there is no different in the OS or local control, pre-operative SRS is associated with lower rates of symptomatic radiation necrosis and LM [54]. Additionally, while relatively low, multiple studies have found that rates of LM were higher with patients treated with SRS compared to WBRT [55,56].

In the recent years, hypofractionated SRT and staged SRS have gained popularity in the management of patients with larger brain metastases with a favorable therapeutic ratio [57,58,59]. Further research will help better define the role of these approaches in the management of brain metastases.

In addition to the cognitive impacts, both SRS and SBRT have the risk of radiation necrosis. The incidence of radiation necrosis in patients with BCBM is unclear but has been reported to be between 5–25% [60]. While most radiation treatment change is transient and asymptomatic, symptomatic radiation necrosis can result in permeant neurologic impairments. For some patients, this permanent neurologic injury can be performance status limiting for future systemic therapies. The dose, volume, fractionation, and prior history of brain radiation are all risk factors for development of RN. At this time, the primary treatment for the management of symptoms related to radiation necrosis is steroids. While largely based on retrospective data, bevacizumab can be used for treatment of radiation necrosis. A meta-analysis of bevacizumab for radiation necrosis included 89 patients and found that 93% had radiographic response and 88% had symptom improvement or resolution [61]. Evaluating the risks of radiation necrosis allows for counseling on the patients individualized risk when considering radiation treatment or systemic treatment options for patients with HER2+ BCBM.

### 4.3. Whole-Brain Radiotherapy

Use of WBRT for treatment of BCBM was established with studies with the median OS of 4–6 months [62,63]. With this dismal prognosis, the cognitive deterioration with WBRT was not initially appreciated. In diseases such as HER2+ BCBM with prolonged survival following the diagnosis of a BM, the neurocognitive and functional impacts of WBRT have led to multiple studies reducing the neurocognitive toxicity. Hippocampal avoidance (HA) as well as memantine are now considered the standard for WBRT when possible, to reduce the neuro cognitive effects [64]. Even with HA and memantine, by 4 months after treatment, over 50% of patients have neurocognitive failure [65]. Given the high risk of neurocognitive failure and the implications on quality of life and performance status, the role of WBRT in patients with HER2+ BCBM should be reconsidered in patients who are potentially candidates for SRS or have systemic therapy options with activity in BCBM.

## 5. Clinical Situations to Consider Delaying Local Therapy

In the treatment of BCBM, there are multiple elements of the clinical situation that may impact treatment decisions for patients that prospective clinical trials are not able to comprehensively evaluate. For all treatment decisions, patients’ clinical performance, prognosis and status of systemic disease are essential considerations. However, for BCBM, the 2022 ASCO guidelines include total number of BCBM, size of the BCBM, as well as the presence or absence of symptomatic mass effect under the consideration for treatment options [6]. The clarification of symptomatic mass effect is important. Patients who, for example, have minimal symptoms such as a seizure or sensory symptom should not be excluded from consideration of upfront systemic therapy given the data for symptomatic benefit to treatment with HER2+ directed systemic therapy. In addition to size and number, the location of the BM in eloquent or high-risk brain regions may impact treatment decisions [66,67,68]. In our experience, however, even for patients with symptoms from high-risk BM such as infratentorial disease or with considerable mass effect can have radiographic and symptomatic improvement (Figure 1) with effective systemic therapy.

While the clarification of symptomatic mass effect is important, it is notable that the ASCO guidelines only factor symptoms into decisions in the setting of newly diagnosed BCBM and not for BCBM that have progressed despite prior local therapy. For progression of BCBM previously treated with WBRT or SRS, trial of systemic therapy is considered for all types of progression, regardless of symptoms or mass effect. Given multiple HER2+ targeted systemic therapies with efficacy in BCBM, we propose a simplified algorithm favoring systemic therapies over local therapies for patients with HER2+ BCBM (Figure 2).

In this proposed algorithm, consideration of systemic therapy must be placed in the context of the current state of systemic disease. With the development of a limited BM in the setting of stable extracranial disease, local therapy is recommended to continue clinical benefit of current systemic therapy. However, when there is development of an extensive BM and the only local therapy available is WBRT, the risks of WBRT must be balanced against the risks of change in systemic agents. In this setting, if one of the agents listed in Table 2 is available for treatment, it is appropriate to consider changing systemic therapy and close observation to delay WBRT. Based on the data for systemic therapies and clinical experience in symptomatic patients, obtaining a follow-up brain MRI 2 months after initiating therapy with BCBM activity is early enough to identify progression despite therapy and WBRT would remain an available option.

For patients with both intracranial and extracranial progression, using symptoms to guide the use of systemic therapy perhaps is more appropriately categorized as symptoms that are likely to improve with local therapy compared to simply identify symptoms as the presence or absence of mass effect. Clinical experience has demonstrated that even symptoms from profound mass effect may improve with use of systemic therapy. This can allow for initiating treatment for both intracranial and extracranial disease without delays for recovery time from surgery or radiation therapy. Notably for both tucatinib and T-DXd, currently there is not safety data to guide timing of radiation with systemic therapy. Similarly, patients with both hormone receptor positive and HER2+, HER2+ targeted therapy are preferred for treatment of the BCBM given limited data for response to hormonal therapy. Ultimately, identification of what symptoms localize to which BCBM and the likelihood of improvement with treatment is best performed in partnership with a brain metastasis clinic or multidisciplinary team composed of medical oncologists, radiation oncologist, oncologic neurosurgeons and if available, neuro-oncologists.

## 6. Future Directions

The above discussion focuses on HER2+ BCBM. However, with the evolving data for T-DXd in patients with low HER-2 expression, the definition of HER2+ BCBM will likely expand from a binary biomarker. Even within the binary definitions for HER2+ expression, type switching does occur between BM and systemic disease. Multicenter work demonstrated 22.8% of patients have biomarker status change from the primary site of the disease to BCBM [69]. In this study, baseline subtype did predict switching with 14.8% of patients who were initially HER2 negative gaining HER2 over-expression or amplification in the BM and that 37.5% of patient with HR+ primary developed discordant subtype metastases. A notable limitation to this work is that it was only able to examine patients who underwent surgical resection. Unfortunately, at this time, there is not a less invasive way to evaluate type switching. For example, in brain progression, serum ctDNA is less commonly detected than systemic progression [70,71]. There is necessary ongoing work to develop less invasive ways to evaluate biomarker status of BCBM. Until such testing is available, short interval MRI imaging can be used to evaluate response to therapy when the systemic therapy is used to treat active BCBM.

As noted above, for patients on tucatinib or T-DXd who need radiation therapy, at this time there is a lack of data regarding safety for the use of radiation with these agents. Data to guide if the therapy needs to be held and the duration of washout before, during and after radiation therapy are of high clinical need. Additionally, further evaluation into early or delayed radiation therapy and impact on survival as well as QOL is urgently needed.

## 7. Conclusions

The hope for prolonged quality of life for patients with HER2+ BCBM has never been more promising. There are now multiple targeted therapies with activity for untreated or progressive BCBM, which is wonderful for patients and challenging for oncologists to make the best choice in what are always unique and complicated scenarios. Shared decision making with patients in treatment planning remains essential, especially now with so many options available. The necessity of multidisciplinary care in the treatment of BCBM remains with the medical oncologist having an increasing role in the treatment of BCBM. More research is needed to identify optimal treatment algorithms and comparison of efficacy of systemic as well as local therapies in patients with BCBM.

## Figures and Tables

**Figure 1 cancers-14-05136-f001:**
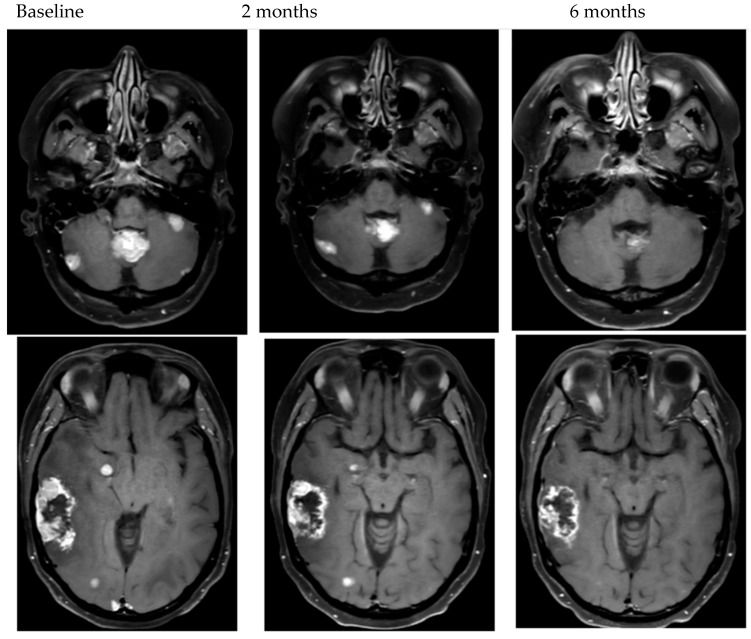
Examples of objective response to HER2-targeted therapy.

**Figure 2 cancers-14-05136-f002:**
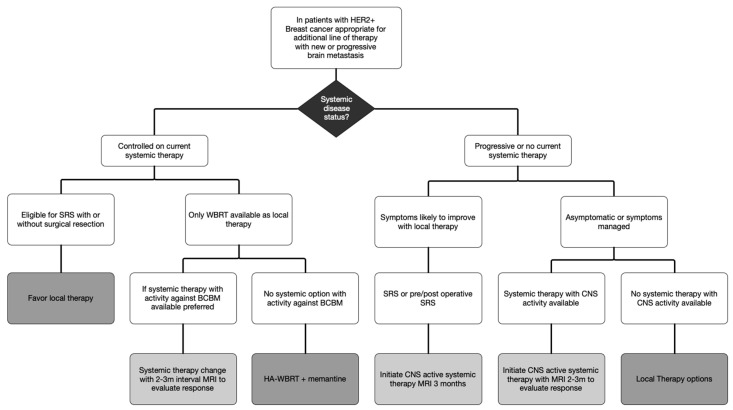
Algorithm for Management of HER2+ Breast Cancer Brain Metastasis.

**Table 1 cancers-14-05136-t001:** Prospective Trials in HER2+ Breast Cancer Brain Metastasis.

Drug	Prospective Trial	Population	N	CNS Response	Survival	Reference
T-DXd	Phase 2 single arm open label: Destiny-Breast01	Treated asymptomatic HER2+ BCBM. Median of 6 lines of prior therapy	24	NR	mPFS 18.1 m	[10]
T-DXd	Phase 3 Destiny-Breast03 T-DXd compared to T-DM1	Treated asymptomatic HER2+ BCBM. Progression on taxane + trastuzumab	82	Median DCR 12.9 m	CNS mPFS: T-DXd-15 m	[11]
T-DM1				Median DCR 7.2 m	TDM1-3 m	
T-DXd	Phase 2 single arm open label: Tuxedo-1 Trial	Untreated or progressing after local therapy HER2+ BCBM	15	Clinical benefit (CR, PR, SD) 92.9% @ 3 m 86.7% @ 6 m CR: 2/15 PR: 9/15	CNS mPFS: 14 m	[12]
T-DXd	Phase 2 cohort open label: DEBBRAH trial	HER2+ treated BCBM, asymptomatic untreated and progressing after local therapy	28	CNS ORR 66.7%	NR–not yet reached	[13]
		Cohort 1—stable after local therapy	8	CNS ORR 80%	CNS PFS @4 m: 87.5%	
		Cohort 2—asymptomatic untreated	11	CNS ORR 50%	NR–not yet reached	
		Cohort 3—progressive after local therapy	9	CNS ORR 44.4%	NR–not yet reached	
tucatinib	Phase 1b with expansion cohort evaluating tucatinib in combination with trastuzumab and capecitabine	HER2+ breast cancer progressive despite TDM1 with and without untreated and progressive after local therapy BCBM. Trial enrolled 60 patients, 33 with BCBM	33	CNS ORR 100%	CNS mPFS 6.7 m mOS: NR	[14]
tucatinib	Phase 3: Trastuzumab + Capecitabine + Tucatinib	HER2+ breast cancer progressive despite TDM1 with and without untreated and progressive after local therapy BCBM.	198	CNS ORR 47.3% DCR @ 3 m 67% DCR @ 6 m 37%	CNS mPFS 9.9 m mOS 18.1 m	[15]
	Herceptin + Capecitabine + placebo		93	CNS ORR-20% DCR @3 m−44% DCR @ 6 m−11.82%	CNS mPFS: 4.2 m mOS-12 m	
TDM1	Phase 3b: KAMILLA	Post hoc analysis of patients with baseline HER2+ BCBM 42.9% no prior radiation	398	CNS ORR 21.4% DCR @3 m -60% DCR @ 6 m -40%	CNS mPFS: 5.5 m OS 18.9 m	[16]
High-Dose trastuzumab (6 mg/kg weekly)	Phase 2: Single arm open label of high-dose trastuzumab with pertuzumab	HER2+ BCBM progressive despite local therapy with stable extracranial disease	39	CNS ORR-11% DCR @ 4 m–68% DCR @6 m–51%	CNS mPFS: 6.6 m mOS: NR	[17]
lapatinib	Phase 2: Lapatinib monotherapy however amended to allow option of lapatinib + capecitabine	HER2+ BCBM progression after radiation therapy	242	CNS ORR (50% or greater)–6% CNS ORR (20% or greater)–21% DCR @2 m–52.5% DCR @4 m–14.7% DCR @6 m−5.9%	CNS mPFS 2.4 m (responders mPFS was 3.38 m) mOS 6.4 m	[18]
lapatinib + capecitabine			50	CNS ORR (50% or greater)–20 (20% or greater) −40% DCR @2 m -66.3% DCR @4 m -37.3% DCR @6 m -19.7%	CNS mPFS: 3.65 m OS NR	
lapatinib + capecitabine	Phase 2: LANDSCAPE Lapatinib + capecitabine	HER2+ BCBM no radiation, lapatinib or capecitabine	45	CNS ORR 65.9% (2 CR, 22 PR) DCR @2 m–78%	CNS mPFS: 5.5 m mOS: 17 m	[19]
neratinib + capecitabine	Phase 2: TBCRC 022	HER2+ BCBM progressive after local therapy	49	CNS ORR 34.2%		[20]
		Lapatinib naïve	37	CNS ORR 49% DCR @3 m−70% DCR@6 m–35%	CNS mPFS: 5.5 m mOS: 13.3	
		Lapatinib treated	12	CNS ORR 33% DCR @3 m–50% DCR@6 m–45%	CNS mPFS: 3.1 m mOS 15.1	
everolimus + lapatinib + capecitabine	Phase 1b/2: TRIO-US-B-09	HER2+ BCBM 63% Prior local therapy 74% previously treated with lapatinib, capecitabine or both	19	CNS ORR 27% DCR @3 m–80% DCR @6 m-57%	CNS mPFS: 6.2 m mOS 24.2 m	[21]
Everolimus + Trastuzumab + Vinorelbine	Phase 2: LCCC 1025	HER2+ BCBM progressive after XRT	32	CNS ORR 4% DCR @3 m −65% DCR @6 m–27%	CNS mPFS: 3.9 m mOS 12.2 m	[22]
Carboplatin + bevacizumab + trastuzumab	Phase 2	HER2+ BCBM new or progressive BM	29	CNS ORR 63% DCR @3 m–79% DCR @6 m–39%	CNS mPFS: 5.62 m mOS 14 m	[23]

BCBM = breast cancer brain metastasis mPFS = median progression free survival, ORR = overall response rate, CNS ORR = overall response rate for brain metastasis, CNS mPFS = median progression free survival of brain metastasis, mOS = median overall survival, CR = complete response, PR = partial response, SD = stable disease, DCR = disease control rate (SD + PR + CR) m = month T-DXd = Trastuzumab deruxtecan, TDM1 = Trastuzumab-emtansine, NR = not reported.

**Table 2 cancers-14-05136-t002:** Preferred systemic agents for patients with HER2+ BCBM.

Tucatinib + Capecitabine + trastuzumab Trastuzumab deruxtecan (T-DXd) Trastuzumab emtansine (T-DM1) High-dose intravenous trastuzumab (6 mg/kg) with pertuzumab

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
