# Peer review of "Management of Brain Metastases from Human Epidermal Growth Factor Receptor 2 Positive (HER2+) Breast Cancer"

_cancers, 2022, doi:10.3390/cancers14205136_

Round 1

Reviewer 1 Report

The formatting of Table 1 is a bit challenging to read. Would consider column lines to demarcate the content of each column. There are also a lot of hyphenated words that should be able to be shifted to allow the entire word to be on one line for ease of reading. 

Would encourage figure one to be formatted all on one page and to have the title at the top of the figure again for ease of reading flow. In the tucatinib section, it would be interesting to add data from TBCRC-049 looking at use of tucatinib for HER2+ leptomeningeal disease as data in this space is so limited. 

In lines 216-221, the grammar can be cleaned up in the description of DESTINY-Breast07 and the description of DESTINY-Breast12 would benefit from some additional detail. In a quick search, it appears this is a Phase 3/4 trial so looking at real-world, postmarketing use. the website also describes additional "efficacy measurements and QOL tools" which would be good to include in this article. 

In lines 324-332 re: radiation necrosis, would consider adding a line or two about utility of bevacizumab in managing RN for completeness. 

In line 340, would clarify the statement about "by 4 months". I think this is intended to mean (by 4 months after treatment".

In line 346, there appears to be an added "this" at the start of the paragraph

In the paragraph following the algorithm and prior to Table 2 as well as in Table 2, there are many providers who would consider tucatinib/cape/trastuzumab as preferred therapy with CNS involvement over T-dxd given the study designs and OS improvement demonstrated in HER2Climb. I would make these interchangeable options and soften the language here demarcating T-dxd as the clear favorite. That would more appropriately reflect NCCN guidelines. I would take a similar approach to discussion of both T-dxd and tucatinib in the first paragraph of the future directions section (lines 397-399)

Line 395 should be deleted. 

Author Response

The formatting of Table 1 is a bit challenging to read. Would consider column lines to demarcate the content of each column. There are also a lot of hyphenated words that should be able to be shifted to allow the entire word to be on one line for ease of reading. Would encourage figure one to be formatted all on one page and to have the title at the top of the figure again for ease of reading flow. 

Thank you for this comment the table formatting has been adjusted to minimize hyphenated words and add column lines. Due to the number of studies and font size limitations unfortunately we are unable  to condense onto a single page. 
In the tucatinib section, it would be interesting to add data from TBCRC-049 looking at use of tucatinib for HER2+ leptomeningeal disease as data in this space is so limited. 

We agree this is important and interesting work. We made the intentional decision to exclude discussion of leptomeningeal metastasis to focus on brain metastasis given the prognostic difference in the discussion of delaying local therapy. We are concerned that discussing the data from TBCRC-049 and other leptomeningeal studies would overwhelm the audience.  

In lines 216-221, the grammar can be cleaned up in the description of DESTINY-Breast07 and the description of DESTINY-Breast12 would benefit from some additional detail. In a quick search, it appears this is a Phase 3/4 trial so looking at real-world, postmarketing use. the website also describes additional "efficacy measurements and QOL tools" which would be good to include in this article. 

Thank you for this comment we have addressed the grammar and added more details regarding these ongoing studies.

In lines 324-332 re: radiation necrosis, would consider adding a line or two about utility of bevacizumab in managing RN for completeness. 

Thank you for this suggestion, we have included discussion and reference regarding role of BEV for radiation necrosis for brain metastasis.  

In line 340, would clarify the statement about "by 4 months". I think this is intended to mean (by 4 months after treatment".

These changes have been made.

In line 346, there appears to be an added "this" at the start of the paragraph

These changes have been made.

In the paragraph following the algorithm and prior to Table 2 as well as in Table 2, there are many providers who would consider tucatinib/cape/trastuzumab as preferred therapy with CNS involvement over T-dxd given the study designs and OS improvement demonstrated in HER2Climb. I would make these interchangeable options and soften the language here demarcating T-dxd as the clear favorite. That would more appropriately reflect NCCN guidelines. I would take a similar approach to discussion of both T-dxd and tucatinib in the first paragraph of the future directions section (lines 397-399)

We appreciate this perspective and agree. The language has been adjusted to have a less biased perspective. We have adjusted the order and removed the ranking in Table 2.  

Line 395 should be deleted. 

These changes have been made.

Reviewer 2 Report

This is a review of HER2+ breast cancer brain metastases treatment in the current era. Given advances in treatment and detection, such a review is an important contribution to the literature. 

This manuscript has several minor editorial issues that should be revised for clarity, and is missing some information about variation in cancer biomarker status.

Specific comments about the manuscript: 

* Line 395 says “section is mandatory, with one or two paragraphs to end the main text”. This seems to be mistakenly included. 

* Line 346 says “This In the treatment of BCBM”, but it seems the word “This” should be removed. 

* Line 406 says “This data”, but “data” are plural.

* Line 190 says “…study reported improvements in ORR 79.7% in patients treated with T-DXd (34.2%) for…”. Is there a missing colon between “ORR” and “79.77%”? 

* Line 166 seems to refer to trastuzumab-capecitabine treatment as a placebo, when it was a control treatment

General comments and recommendations: 

* Please address limitations or acknowledge additional future directions for the proposed algorithm. For example: 

* The proposed algorithm assumes HER2 status is binary. However, the authors cite a trial about treatment of disease with low levels of HER2+ (IHC1+ or IHC2+ with negative FISH). If trials are conducted where HER2 status is more than a binary value, it seems important to address heterogeneity in HER2+ disease. Please address how the continuous nature of HER2 status could affect treatment decisions. 

* Should there be no difference in treatment decisions for patients who have hormone receptor positive versus negative disease? Whether hypothetical or evidence-based, please comment on how or whether hormone receptor status could affect decisions within the proposed algorithm. 

* The authors recognize that subtype switching occurs. However, there is no suggestion about how this may affect the decision to pursue local versus systemic management. This comment is made in the future directions section, but the authors’ hypotheses about such contexts seems valuable given their work in this article. 

Author Response

This is a review of HER2+ breast cancer brain metastases treatment in the current era. Given advances in treatment and detection, such a review is an important contribution to the literature. 

The authors thank the reviewer for the kind comments.

This manuscript has several minor editorial issues that should be revised for clarity, and is missing some information about variation in cancer biomarker status.

Specific comments about the manuscript: 

  • Line 395 says “section is mandatory, with one or two paragraphs to end the main text”. This seems to be mistakenly included.

These changes have been made.

  • Line 346 says “This In the treatment of BCBM”, but it seems the word “This” should be removed.

These changes have been made.

  • Line 406 says “This data”, but “data” are plural.

These changes have been made.

  • Line 190 says “…study reported improvements in ORR 79.7% in patients treated with T-DXd (34.2%) for…”. Is there a missing colon between “ORR” and “79.77%”?

These changes have been made.

* Line 166 seems to refer to trastuzumab-capecitabine treatment as a placebo, when it was a control treatment

These changes have been made.

General comments and recommendations: 

* Please address limitations or acknowledge additional future directions for the proposed algorithm. For example: 

  • The proposed algorithm assumes HER2 status is binary. However, the authors cite a trial about treatment of disease with low levels of HER2+ (IHC1+ or IHC2+ with negative FISH). If trials are conducted where HER2 status is more than a binary value, it seems important to address heterogeneity in HER2+ disease. Please address how the continuous nature of HER2 status could affect treatment decisions.

With excellent suggestion to further highlight the evolving definition of HER2 status and  type switching we have adjusted the future direction section to highlight the limitations.

  • The authors recognize that subtype switching occurs. However, there is no suggestion about how this may affect the decision to pursue local versus systemic management. This comment is made in the future directions section, but the authors’ hypotheses about such contexts seems valuable given their work in this article.

With excellent suggestion to further highlight the evolving definition of HER2 status and  type switching we have adjusted the future direction section to highlight the limitations.

  • Should there be no difference in treatment decisions for patients who have hormone receptor positive versus negative disease? Whether hypothetical or evidence-based, please comment on how or whether hormone receptor status could affect decisions within the proposed algorithm.

Thank you for highlighting this important limitation in the literature. We have included the fact that the algorithm is for HER2+ disease regardless of hormone receptor status.